



# Comparison of measured and simulated fatigue loads on a multi-megawatt wind turbine

Ansh Patel[1,2], Jakob Mann[1], Mikael Sjöholm[1], Kasper Zinck[2], and Karunya Raj[2]

[1]DTU Wind and Energy Systems, Technical University of Denmark Risø Campus, 4000 Roskilde, Denmark
[2]Vestas Wind Systems A/S, Hedeager 42, 8200 Aarhus, Denmark

**Correspondence:** Ansh Patel (patel@dtu.dk)

**Abstract.** The Mann turbulence model is widely used in the design and certification of multi-megawatt wind turbines. However, these turbines operate in a region of the atmosphere where the model's assumptions are violated. One of the most significant assumptions is that of neutral stability conditions, which raises concerns about the model's accuracy for load simulations. To investigate this, we compare fatigue loads measured on a 15 MW wind turbine with simulations performed using an aeroelastic solver. The inflow was characterized using data from a meteorological mast equipped with sonic and cup anemometers. The turbulence model was fitted to measurements of auto-spectra under varying wind speeds and stability conditions, while the vertical profile of wind speed was represented by a power law. The resulting wind fields were then used as input to the aeroelastic simulations.

We first present a comparison of measured fatigue loads on the tower and blades across different atmospheric stability regimes. The difference in loads between unstable and stable conditions was found to be 98% for the tower and 20% for the blades, underscoring the importance of accounting for atmospheric stability in wind turbine siting and load verification campaigns. A subsequent comparison of measurements and simulations revealed that loads from the two sources tend to fall within three standard deviations of each other, even under non-neutral stability conditions. However, the simulated fatigue loads on the tower were overestimated by a margin of five standard deviations under some stable conditions, likely due to incorrect predictions of the spectral coherence made by the turbulence model. Shear extrapolation based on the power law might also lead to overestimation of blade loads in the simulations.

These results indicate that, despite its simplifying assumptions, the Mann model when fitted to measurements of turbulence auto-spectra, does not introduce significant errors in fatigue load simulations for solitary multi-megawatt wind turbines.

## 1 Introduction

The Wind turbines - Part 1: Design requirements of the International Electrotechnical Commission (IEC) (IEC, 2019) recommends the Mann (Mann, 1994) and Kaimal (Kaimal et al., 1972) spectral turbulence models as inputs for aeroelastic simulations of wind turbines. These models are designed to capture turbulence in the surface layer of the atmosphere. However, modern wind turbines with rotor diameters greater than 200 m and hub heights exceeding 100 m often operate outside the surface layer. The assumptions behind the aforementioned models are no longer valid in this region of the atmosphere (Veers et al., 2023).



For instance, the models assume neutral stability conditions, homogeneous turbulence in all directions of space and Gaussian statistics at the smallest scales. On the contrary, atmospheric turbulence is often thermally generated, is inhomogeneous in the vertical direction and follows non-Gaussian statistics at the smallest scales. Thus, it is uncertain whether the Mann and Kaimal models can provide accurate estimates of the dynamic loads on multi-megawatt wind turbines.

The aim of this study is to gauge the accuracy of load simulations where the input turbulence field is described by the Mann

model, by comparing with measurements of fatigue loads from a 15 MW prototype wind turbine.

The impact of inflow turbulence on the fatigue loads experienced by a wind turbine has been well documented in the literature. Robertson et al. (2019) performed a sensitivity study to evaluate the influence of various inflow parameters such as turbulence intensity, length scales, vertical shear and veer. They found that the turbulence and length scale of the streamwise wind component, along with vertical shear, played a primary role in determining the fatigue loads on a turbine. Sathe et al.

(2013) theoretically studied the impact of atmospheric stability on fatigue loads. They showed that the tower and rotor loads were significantly higher during unstable conditions compared to the loads under stable stratification. Nybø et al. (2021a) simulated the fatigue loads on a 15 MW wind turbine using different inflow specifications such as turbulence boxes generated using the Mann and Kaimal models and wind fields from Large Eddy Simulations (LES). The simulated loads showed considerable differences even when the three methods had the same one-point statistics. The variability was attributed to the difference in

the spatial structure of turbulence, but given the lack of loads measurements, it was not possible to determine which method was the most accurate.

Concurrently, the Mann model has been modified to capture phenomena that were lacking in the original formulation. These include atmospheric stability (Chougule et al., 2018), dependence on time (de Maré and Mann, 2016), mesoscale turbulence (Syed and Mann, 2024) and inhomogeneity (Guo et al., 2026). On the other hand, non-Gaussian turbulence and effects of

the Coriolis force have not been included in the model as of yet. However, given the absence of measurements of loads from an operational wind turbine to serve as the ground truth, it is unclear whether these phenomena lead to significant errors in aeroelastic simulations. Resolving this uncertainty requires a so-called "one-to-one" comparison, where measured loads are compared to simulated loads. The actual inflow experienced by the turbine is replicated in the simulation by using measurements of turbulence from a met-mast or a lidar. In this regard, Brown et al. (2024) performed a one-to-one comparison on a

2.8 MW turbine. Inflow measurements from an array of sonic anemometers, wind vanes, and a nacelle lidar were assimilated into the simulation environment using a few different methods (Jonkman and Buhl, 2006; Rinker, 2022). This study showcased the value of one-to-one comparisons while also highlighting the associated difficulties. The flapwise blade and fore-aft tower loads from the simulations showed a significant bias compared to the measurements due to inaccuracies in the aerodynamic modelling and the absence of certain controller features in the simulation environment. As the blade design and controller

architecture of a commercial wind turbine are confidential, it is difficult to replicate the turbine within a simulation. However, the work of Dimitrov et al. (2024) did not suffer from these drawbacks. They compared inflow assimilation using met-masts to lidar-based methods and found that the uncertainties in the loads estimates from the two methods were similar and occasionally lower when using the lidar. Moreover, the simulated loads agreed with the measurements at different wind speeds and turbulence intensities. However, this study considered an older turbine variant with all measurements taken below 100 m in





the atmosphere. Thus, the validity of the Mann model for load simulations of multi-megawatt wind turbines remained an open question.

We present fatigue loads measured on a 15 MW prototype under varying atmospheric conditions and compare them to aeroelastic simulations with the same controller architecture, aerodynamic, and structural properties as the real turbine. The inflow is assimilated by fitting the Mann model to measurements of turbulence auto-spectra from a 155 m tall met-mast. To the authors' knowledge, the relationship between fatigue loads on a wind turbine operating in the field and atmospheric stability has previously not been documented in the literature.

The paper is organised as follows: Sect. 2 presents the measurement campaign, the method of inflow assimilation and a brief overview of the simulation environment. The results are described in Sect. 3 starting with the dependence of fatigue loads on atmospheric stability followed by one-to-one comparisons of blade and tower loads. Finally, the conclusions of this study are reflected upon in Sect. 4.

## 2 Methods

### 2.1 Measurement campaign

The purpose of the measurement campaign was, among others, power curve verification and load validation of the Vestas V236 prototype wind turbine. The turbine has a rotor diameter of 236 m and a hub-height of 163 m while the rated power output is 15 MW. It is an offshore wind turbine and several variants are currently under installation in wind farms in the North Sea and the Baltic Sea.

The campaign lasted for two years, from 2023 to 2025 and took place at Test Center Østerild. It is located in north Jutland, Denmark about 50 km from the coast. We refer the reader to Peña (2019) for a detailed description of the site. The prototype turbine was equipped with many sensors that measured bending moments on various components, the power output of the turbine, pitch angle of all the blades, rotation speed etcetera. An overview of the signals used in this study is shown in Table 1. An IEC-complaint meteorological mast was also present about 660 m to the west of the prototype. Since the dominant wind direction was westerly, the mast was often present upstream of the turbine. It was equipped with a sonic anemometer at 155 m, cup anemometers at 45 m, 104 m and 163 m, and wind vanes at 45 m and 159 m above ground level.

### 2.2 Processing and binning of data

The data analysed in this study consists of a subset of the data collected during the campaign obtained after the filtering steps described below. The filtered dataset was subsequently binned according to the mean wind speed and atmospheric stability. The data within each bin was further filtered to obtain turbulence measurements appropriate for a one-to-one comparison. The details of data processing are described below.

The data were sampled at a frequency of 5 Hz and partitioned into periods of 10 minutes. Only those periods were selected for further analysis, where the mean wind direction $\bar{\theta}$ measured by the wind vane at 163 m was between 225° to 315°.



| Signal (notation) | Measurement device |
|---|---|
| Power output (P) | Converter |
| Rotation speed of low speed shaft ($\Omega$) | Tachometer |
| Pitch of $i$-th blade ($\zeta_i$) | Pitch sensor |
| Bending moment at root of $i$-th blade in flapwise direction ($M_{bi}^{f}$) | Strain gauge |
| Bending moment at root of $i$-th blade in edgewise direction ($M_{bi}^{e}$) | Strain gauge |
| Bending moment at tower bottom in mean wind direction ($M_{tb}^{fa}$) | Strain gauge |
| Bending moment at tower bottom in direction perpendicular to mean wind ($M_{tb}^{ss}$) | Strain gauge |
| Bending moment at tower top in direction perpendicular to mean wind ($M_{tt}^{fa}$) | Strain gauge |
| Bending moment at tower top in direction perpendicular to mean wind ($M_{tt}^{ss}$) | Strain gauge |

**Table 1.** An overview of the signals analysed in this study and the devices that measure them.

Note that $\bar{\cdot}$ refers to time averaging and the wind directions are expressed in meteorological convention with 0° being wind coming from the north. The westerly sector is selected because winds from this direction can be assumed to be homogenous and free from the wake of neighbouring turbines (Peña, 2019). Subsequently, the 10 min mean wind speed ($\overline{u}$) and direction measured by the sonic anemometer were compared to the cup and vane. The purpose of this comparison was to identify inaccuracies in the instruments and remove the corresponding periods from the dataset. As seen in Fig. 1, the instruments agreed in the measurements of the mean wind speed and direction. Although biases of 0.3 ms⁻¹ and 1° are detected in (a) and (b) respectively, this can be attributed to the sonic anemometer being placed 8 m and 4 m lower than the cup and vane respectively. Next, the 10 min periods where the mean wind speed was outside the turbine operating region were removed. This was followed by calculating the Monin-Obukhov length ($L_{mo}$) for each of the remaining 10 min periods:

$$L_{mo} = \frac{-u_*^3 T}{\kappa g \overline{w'T'}}, \tag{1}$$

where $T$ is the absolute temperature, $\kappa$ is the von Kármán constant, $w'$ is the fluctuation in the vertical-wind component and $\overline{w'T'}$ is the vertical temperature flux. Moreover, the friction velocity, $u_*$ is given by:

$$u_* = \sqrt{-\overline{uw}}. \tag{2}$$

The 10 minute data were subsequently binned according to the mean wind speed and the Monin-Obukhov length based on the stability classifications suggested by Gryning et al. (2007) and shown in Table 2. The size of the mean wind speed bins was 2 ms⁻¹ with the bin centres being odd numbers from 5 to 21.

The probability distribution of $L_{mo}$ conditioned on $\overline{u}$ is shown in Fig. 2. We found stable conditions to be most frequent at low wind speeds, contrary to the observations of Peña (2019) wherein unstable conditions were dominant at wind speeds below 11 ms⁻¹. This observation is likely a sampling bias. The instruments briefly stopped recording data in the summer of 2024 when unstable conditions are more frequent. At mean wind speeds above 14 ms⁻¹, mechanically generated turbulence is stronger than the effects of buoyancy. Consequently, neutral conditions were frequently observed at higher wind speeds.

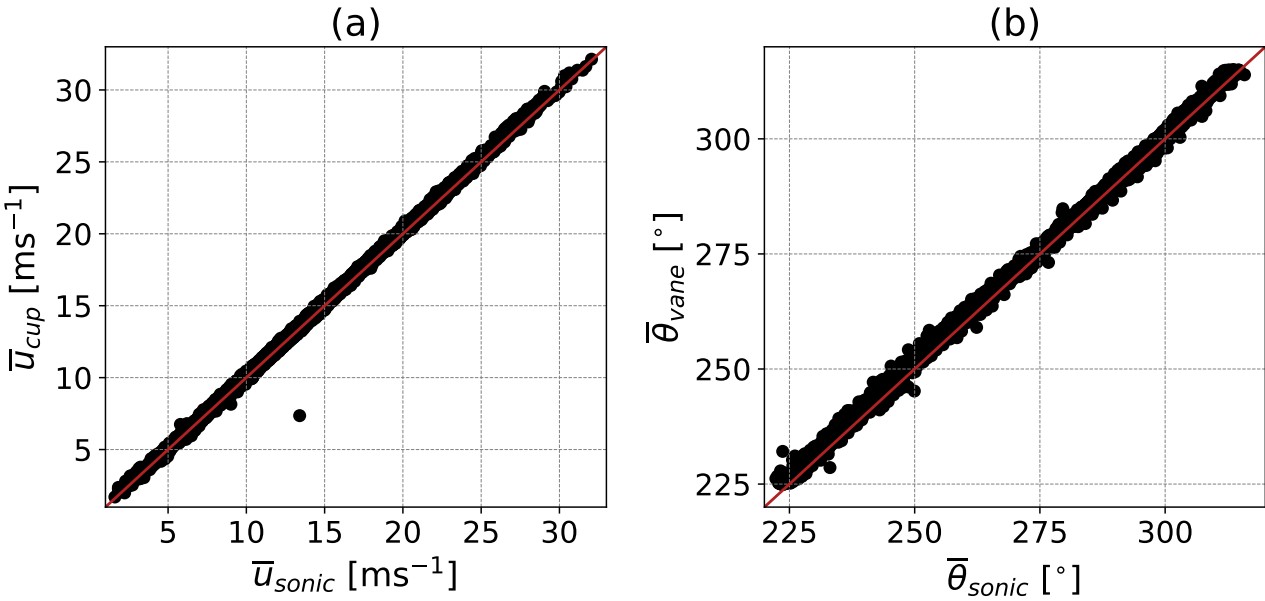

**Figure 1.** (a): The 10 min mean wind speed measured by the sonic anemometer at a height of 155 m compared to measurements made by the cup anemometer at a height of 163 m, when the mean wind direction was between $225°$ to $315°$. (b): The 10 min mean wind direction (in meteorological coordinates) measured by the sonic anemometer compared to measurements made by the wind vane at a height of 159 m. Note that the red line has a slope equal to one.

| Stability | Monin-Obukhov length |
|---|---|
| Very stable (vs) | $10 \leq L_{mo} < 50$ |
| Stable (s) | $50 \leq L_{mo} < 200$ |
| Near neutral stable (nns) | $200 \leq L_{mo} \leq 500$ |
| Neutral (n) | $|L_{mo}| > 500$ |
| Near neutral unstable (nnu) | $-500 \leq L_{mo} < -200$ |
| Unstable (u) | $-200 \leq L_{mo} < -100$ |
| Very unstable (vu) | $-100 \leq L_{mo} \leq -50$ |

**Table 2.** Classifications of atmospheric stability based on the Monin-Obukhov length.





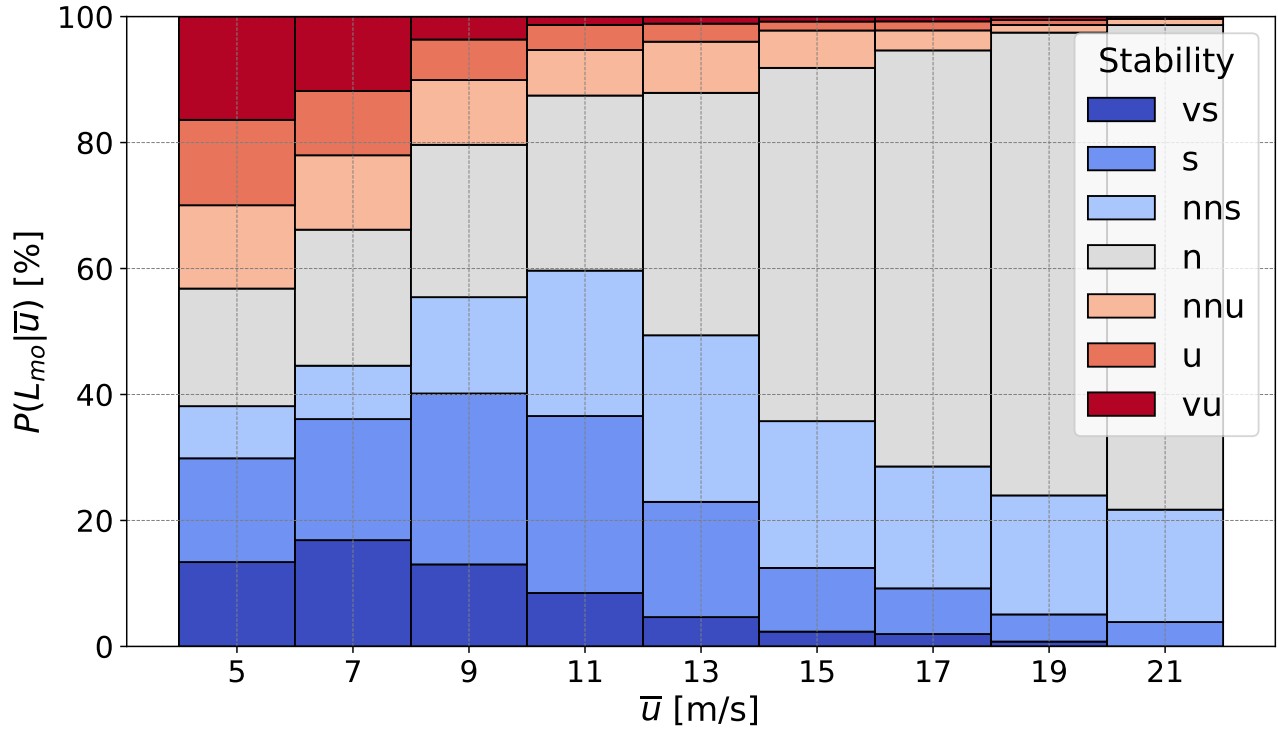

**Figure 2.** The distribution of atmospheric stability quantified by the Monin-Obukhov length ($L_{mo}$) conditioned on the 10 min mean wind speed ($\overline{u}$).

In the subsequent processing step, we aimed to find the longest period of contiguous measurements within each bin. Table 3 shows the capture matrix indicating the number of contiguous 10 min periods ($N_{bin}$) found for each bin. The turbulence measurements (namely the auto-spectra) were averaged over $N_{bin}$ periods and fitted to the Mann model. If $N_{bin}$ was less than

115 four, the uncertainties in the measurements were deemed to be too high and a fit with the Mann model could not be obtained. As a consequence, a one-to-one comparison could not be carried out in the corresponding bins. Hence, there were a total of 39 bins with sufficient amount of data.

The choice to limit the analysis to periods with contiguous data was motivated by the observation that wind speed and stability did not fully describe turbulence at these heights. Indeed, when computing the spectra for all 10 min periods within

120 a given bin, we found the spread to be large, encompassing at least three decades. Additionally, as described later, spectral fitting was the preferred method for inflow assimilation. Thus, by attempting to fit the Mann model to the bin-averaged spectra, we risked lumping together distinct atmospheric conditions which would add significant uncertainty to the comparisons of loads. On the other hand, by selecting a contiguous period of measurements with minimal spread in the individual spectra, the resultant uncertainty was lower and easier to quantify as it was purely statistical in nature.





| $\overline{u}$-bin centre [ms$^{-1}$] | vs | s | nns | n | nnu | u | vu |
|---|---|---|---|---|---|---|---|
| 5 | 6 | 4 | - | 4 | 5 | 4 | 6 |
| 7 | 8 | 7 | - | 9 | 4 | 4 | - |
| 9 | 9 | 17 | 6 | 10 | 6 | 4 | - |
| 11 | 6 | 11 | 8 | 14 | 5 | 4 | - |
| 13 | 4 | 7 | 9 | 10 | 5 | - | - |
| 15 | - | 6 | 7 | 10 | - | - | - |
| 17 | - | 7 | 4 | 16 | - | - | - |
| 19 | - | 5 | 10 | 12 | - | - | - |
| 21 | - | - | 5 | 10 | - | - | - |

**Table 3.** The number of contiguous 10 min periods per mean wind speed and stability bin used to compute the auto-spectra of turbulence.

### 2.3 Assimilation of inflow

The goal of inflow assimilation is to achieve a faithful representation of the measured inflow in the simulation environment. To replicate the turbulence in the simulations, we use spectral fitting. Herein, the Mann model is fitted to the auto-spectra of the along-wind $u$, cross-wind $v$ and vertical $w$ wind components as well as the real part of the $uw$-cross-spectrum to obtain the three model parameters: $L$, $\alpha\varepsilon^{2/3}$ and $\Gamma$ for each stability and wind speed bin. These parameters are then used to generate a turbulence box which is provided as input to the aeroelastic solver (Mann, 1998).

Previous one-to-one comparisons have used different inflow assimilation methods such as scaling a turbulence box with the measured variances in the three components or by constraining a turbulence box on a set of measurements. However, the former has been shown to possess a high level of uncertainty (Liew and Larsen, 2022) while the latter is more suitable for measurements from a nacelle lidar (Rinker, 2022). Since measurements of the auto-spectra contain the length scale and variance, both of which have a significant influence on fatigue loads, fitting the turbulence model to the spectra can achieve accurate assimilation of the inflow for the purposes of load validation (Nybø et al., 2021b).

The fitting procedure is as follows. Measurements of auto-spectra are computed by:

$$S_i(f) = \frac{2\pi}{N_{bin}T} \sum_{k=1}^{N_{bin}} \left| \mathrm{u}_i^k(f,T) \right|^2, \tag{3}$$

where,

$$\mathrm{u}_i^k(f,T) = \frac{1}{2\pi} \int_0^T u_i^k(t)\exp(-2\pi ift)\mathrm{d}t, \tag{4}$$

is the Fourier transform of the $i$-th component such that $(u_1, u_2, u_3) = (u, v, w)$ and $T = 600$ s. Note that measurements of the three wind components at a sampling frequency of 5 Hz are obtained from the sonic anemometer. The spectra are further block-averaged over logarithmic bins, $N_l$, to reduce the level of noise at the high frequency end of the spectrum. The block-averaging is done such that there are 12 points per decade which means that $N_l$ increases with frequency. Hence, the statistical





uncertainty in the measurements, $\sigma(\langle S_i \rangle)$ is quantified by:

$$\sigma(\langle S_i \rangle) = \frac{\langle S_i \rangle}{\sqrt{N_{bin} N_l}}. \tag{5}$$

Then, the model parameters which minimize the error, $\xi^2$, are obtained by an optimization algorithm. $\xi^2$ is defined as:

$$\xi^2 \left( L, \Gamma, \alpha\varepsilon^{\frac{2}{3}} \right) = \sum_{i=1}^{3} \sum_{n=1}^{N} \left[ \log\left(k_1^n F_i\right) - \log\left(k_1^n F_{i,m}\right) \right]^2 + \sum_{n=1}^{N} \left[ \log\left(k_1^n \Re(\chi_{uw})\right) - \log\left(k_1^n \Re(\chi_{uw,m})\right) \right]^2, \tag{6}$$

where the subscript $m$ refers to the measured spectra and $\chi_{uw}$ is the $uw$ cross-spectrum. On the other hand, the modelled
spectra are given by:

$$F_i \left( k_1; L, \Gamma, \alpha\varepsilon^{\frac{2}{3}} \right) = \iint\limits_{-\infty}^{\infty} \Phi_{ii} \left( \boldsymbol{k}; L, \Gamma, \alpha\varepsilon^{\frac{2}{3}} \right) \mathrm{d}k_2 \mathrm{d}k_3 \ , \tag{7}$$

Here, $\boldsymbol{\Phi}(\boldsymbol{k})$ is the spectral tensor derived by Mann (1994) defined over wave number space: $\boldsymbol{k} = (k_1, k_2, k_3)$. Note that Taylor's
hypothesis is used to move between frequency and wave number space.

Figure 3 shows the measured spectra under different stability conditions along with the fit derived from Eq. (6) at a mean
wind speed of 9 ms$^{-1}$. Note that the area under the spectra of a given component is equal to the variance of that component
while the location of the spectral peak is related to the length scale. As excepted, unstable conditions show much more variance
(or turbulence intensity) and larger length scales as compared to stable conditions while neutral conditions are intermediate.
The Mann model fits the spectra well in the given frequency range. The model parameters obtained in each stability and wind
speed bin are shown in Fig. 4. The parameter $\alpha\varepsilon^{2/3}$ is proportional to the square of the mean wind speed (Syed and Mann,
2024) and hence $\sqrt{\alpha\varepsilon^{2/3}}$ is observed to vary linearly with $\overline{u}$. Furthermore, $\alpha\varepsilon^{2/3}$ is also related to the variance in the wind
speed (Kelly, 2018). Thus, Fig. 4 (a) shows that higher wind speeds are associated with more turbulence. Length scales as large
as 180 m are observed in certain unstable conditions while stable conditions usually show length scales lower than 50 m. The
anisotropy parameter, $\Gamma$, does not show any appreciable dependence on the mean wind speed or stability and tends to be close
to 3 in most cases.

Given the importance of vertical shear and veer on fatigue loads (Robertson et al., 2019), we also attempted to replicate the
measured variation of mean wind speed and direction with height in the simulation setup. Data from three cup anemometers
present at heights 45 m, 105 m and 163 m and two wind vanes present at 45 m and 159 m were used to obtain the mean vertical
profiles of the along-wind component and the wind direction: $\overline{u}(z)$ and $\overline{\theta}(z)$. Next, the shear parameter, $\alpha_s$, was extracted by
fitting the power law profile to the measurements:

$$\overline{u}(z) = \overline{u}(z_h) \left( \frac{z}{z_h} \right)^{\alpha_s}, \tag{8}$$

where $z_h$ is the hub-height. Here, the shear above the rotor is assumed to follow the same power profile as measured below the
rotor. The veer profile, on the other hand, is assumed to be a linear function of height:

$$\overline{\theta}(z) = \beta(z - z_h). \tag{9}$$



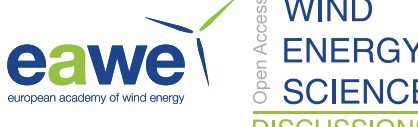

**Figure 3.** The measured, pre-multiplied auto-spectra of the $u$, $v$ and $w$ components and the real part of the $uw$ cross-spectrum under varying stability conditions and a mean wind speed of 9 ms$^{-1}$.





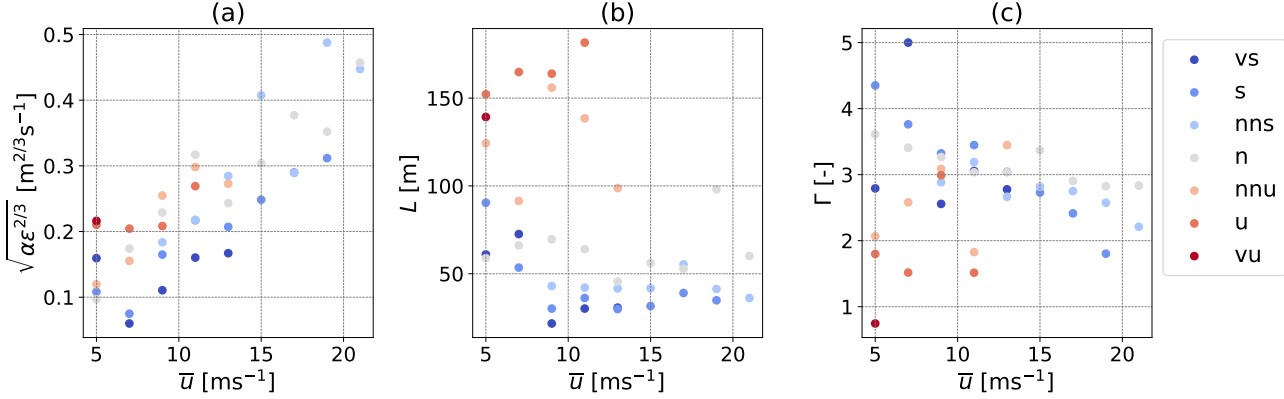

**Figure 4.** Parameters of the Mann turbulence model as functions of wind speed and stability obtained by fitting the model to the measurements shown in Table 3.

$\beta$ is obtained by fitting the measured profile of $\overline{\theta}(z)$ to Eq. (9). The values of $\alpha_s$ and $\beta$ for all stability and wind speed bins are

shown in Fig. 5. A larger variation with height in the wind speed and direction is observed during stable conditions as compared to unstable conditions as more turbulent mixing occurs in the latter which leads to more uniformity in the mean wind speed and direction.

Thus, the wind box, $\boldsymbol{u_p}$, provided to the aeroelastic solver is:

$$\boldsymbol{u_p}(x,y,z) = \boldsymbol{u_m}(x,y,z;\alpha\varepsilon^{2/3},L,\Gamma) + \boldsymbol{e_x}\overline{u}(z_h)\left(\frac{z}{z_h}\right)^{\alpha_s} + \boldsymbol{e_y}\overline{u}(z)\tan(\beta(z-z_h)). \tag{10}$$

Here, $\boldsymbol{u_m}$ is the wind box of turbulent fluctuations with zero mean generated using the Hipersim library (Dimitrov et al., 2024) and $\boldsymbol{e_x}$, $\boldsymbol{e_y}$ are the unit vectors in the along-wind and cross-wind directions.

The turbulence boxes had a temporal resolution of $\Delta t = 0.15$ s, hence the spatial resolution in the along-wind direction, $\Delta x$, was $\overline{u}(z_h)\Delta t$. Given a total simulation time of 600 s, the number of grid points in the $x$-direction were 8094. On the other hand, the vertical and lateral grid resolution was 3.8 m with 64 grid points in either direction. These values were recommended

by Liew and Larsen (2022) for the convergence of fatigue load estimates. They also suggest using at least seven turbulence seeds to reach a standard error of less than 5% in the damage equivalent moment on the blades and the tower. Thus, in this study, a total of $7 \times 39 = 273$ simulations were performed.

## 2.4 A note on the aeroelastic solver

The dynamic response of the turbine to a given wind input is simulated in Vestas turbine simulator (VTS). It is a state of

the art blade element and momentum theory (BEM) solver derived from Flex5 (Øye, 1996) and based on the mode shape approach using generalised coordinates, masses, stiffnesses and forces. The solver is non-linear as mass, damping and stiffness matrices are recalculated at every time step. It includes tip loss correction, a dynamic stall model and is linked to the controller





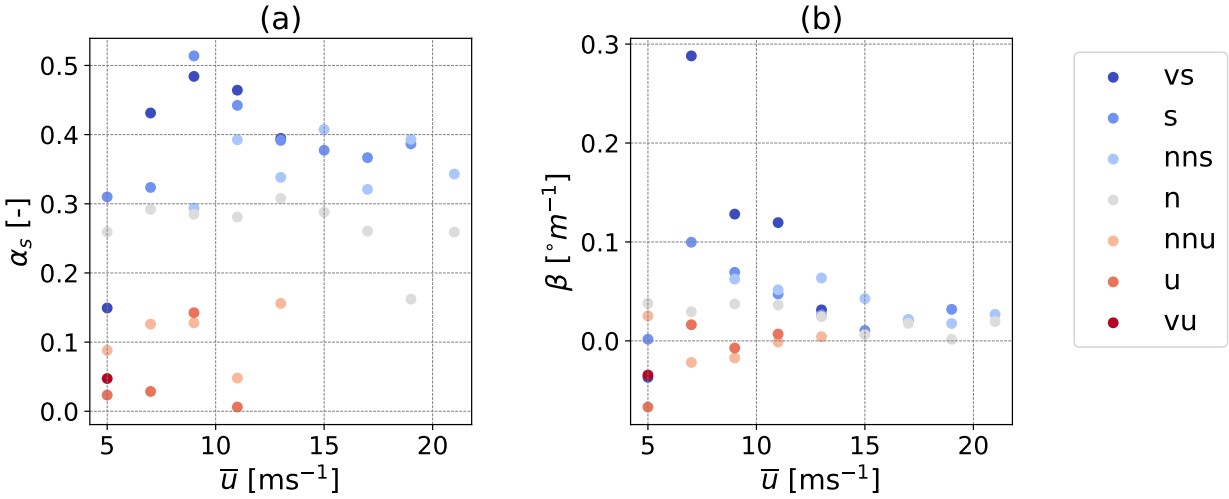

**Figure 5.** The shear and veer parameters as functions of wind speed and stability obtained by fitting Eq. (8) and (9) to measurements of the vertical profiles of mean wind speed and direction.

via a dynamic-link library (DLL). Since the system of equations is not stiff, a very efficient equation solver may be used (Runge-Kutta-Nyström with Cholesky decomposition), with a relatively large constant time step.

## 3 Results

This section presents an analysis of the fatigue loads measured on the prototype turbine which are further compared to the simulation results from the aeroelastic solver. Fatigue loads are often characterized by the 1 Hz damage equivalent moment $D_M$ which is based on the Palmgren-Miner damage rule (Burton et al., 2011). Consider $S(t)$ to be the time-dependent bending moment on a particular component. $S(t)$ is first decomposed into a set of single amplitude load ranges, $S_i$, and the number of cycles for each load range, $n_i$, are calculated using rain-flow counting (Rychlik, 1987). Moreover, the number of cycles to failure, $N_i$, at a certain load range are found using the Wöhler S-N curve:

$$N_i = \left( \frac{S_0}{S_i} \right)^m, \tag{11}$$

where $m$ is the Wöhler exponent of the material and $S_0$ is the $y$-intercept of the S-N curve. Note that in this study, $m$ is taken as 4 for steel and 10 for glass fibre composite (Slot et al., 2019). The fatigue damage suffered by the material at particular load range is given by:

$$D_i = \frac{n_i}{N_i}. \tag{12}$$





Thus, the total damage according to Palmgren-Miner's rule is:

$$D = \sum D_i = \sum \frac{n_i}{N_i}. \tag{13}$$

Finally, $D_M$ is defined as the single amplitude time-varying load applied on the component for $n_{eq}$ cycles such that it results
in the same level of accumulated damage as $S(t)$:

$$D_M(S) = \left( \frac{\sum n_i S_i^m}{n_{eq}} \right)^{\frac{1}{m}}, \tag{14}$$

where $n_{eq}$ is 600 which implies that $D_M(S)$ is applied at a frequency of 1 Hz for a duration of 10 minutes.

We also remind the reader that all the results presented in these sections have been anonymized. This was achieved by scaling
the absolute value of a given quantity (say $X$) as follows:

$$\tilde{X} = \frac{X}{X_{max} - X_{min}}, \tag{15}$$

where $X_{min}$ and $X_{max}$ were the minimum and maximum values of $X$ respectively. Note that the minimum and maximum are
the same for a given load channel and are computed over all stabilities and mean wind speeds. Thus, all loads from a particular
channel, irrespective of whether they are obtained from measurements or simulations, are scaled with the same values of $X_{min}$
and $X_{max}$.

## 3.1  Effect of atmospheric stability on fatigue loads

The mean damage equivalent moments are computed from the measurements of bending moments at the tower-bottom and
blade root and are shown in Fig. 6 and Fig. 7 as functions of wind speed and atmospheric stability. Note that the fore-aft
direction is parallel to the mean wind direction and side-side is perpendicular to it. Moreover, Fig. 7 shows the damage equiv-
alent loads on one of the blades although similar results were obtained for all three. Lastly, the error bar indicates the standard
deviation of the damage equivalent moment, which is also scaled according to Eq. (15).

Figure 6 (a) shows that the damage equivalent moments at the tower-bottom in the fore-aft direction are significantly influ-
enced by atmospheric stability. For instance, in the 11 ms$^{-1}$ wind speed bin, which straddles the turbine's rated wind speed,
the fatigue loads are twice as high under unstable and neutral conditions as compared to stable conditions. Averaging over
the wind speed bins where data is simultaneously available for unstable, neutral and stable conditions, the damage equivalent
moments are 98% and 65% higher under unstable and neutral conditions as compared to stable ones. Moreover, the standard
deviation of the damage equivalent moment is also 1.5 times higher under unstable and neutral conditions. On the other hand,
the fatigue loads in the side-side direction are not as strongly influenced by stability. The difference in damage equivalent loads
in this direction between unstable and stable conditions is 17%. Similar trends are observed in the damage equivalent moments
at the tower-top.
As seen in Fig. 3 and Fig. 5, unstable conditions are generally associated with higher turbulence intensities and larger length
scales but lower shear in the vertical wind profile. Thus, the results presented herewith show that tower loads are mainly driven

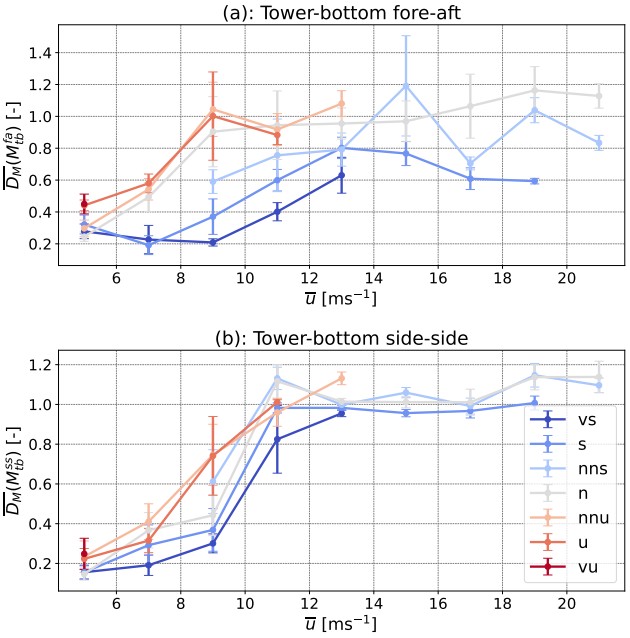

**Figure 6.** The mean damage equivalent moments on the tower-bottom in the (a) fore-aft and (b) side-side directions as functions of wind speed and stability. Note that the data has been anonymized using Eq. (15) and the error bars indicate the standard deviation. Moreover, (a) and (b) are scaled with different values of $X_{min}$ and $X_{max}$.

by turbulence and not shear. Indeed, it is expected that a three-bladed rotor will average out the effects of load variation due to the wind profile. The turbulence intensity ($TI$), on the other hand, quantifies the fluctuations of a given wind components around its mean and consequently, a larger $TI$ of the along wind component can lead to larger variations in tower loads.

However, the $TI$ alone is not a sufficient proxy for fatigue loads. The length scale describes the size of the eddies which drive the turbulent fluctuations. Eddies which are comparable to the rotor diameter cause variations in the dynamic loads while eddies which are smaller in size are filtered out. Hence, the $TI$ and length scale together are responsible for fatigue loads on the tower being 98% higher under unstable conditions as compared to stable conditions. We also note that the damage equivalent moments under neutral conditions do not necessarily represent the mean loads at any given wind speed. They appear to be

closer to unstable conditions than to stable ones. This is because the $TI$ and length scale of the load driving $u$-component under neutral conditions was closer to the values under unstable conditions.

The damage equivalent loads on the blade in the flapwise direction are impacted to a lower degree by atmospheric stability. At wind speeds below rated, unstable and neutral conditions cause approximately 20% more fatigue damage as compared to stable conditions. However, at wind speeds above rated, the damage equivalent loads under neutral and stable conditions differ

by -2%. These observations are caused by the opposing effects of turbulence and shear, both of which affect the fatigue loads on a wind turbine blade. While unstable conditions are more turbulent, the mean wind speed does not vary appreciably with



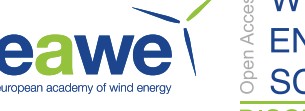

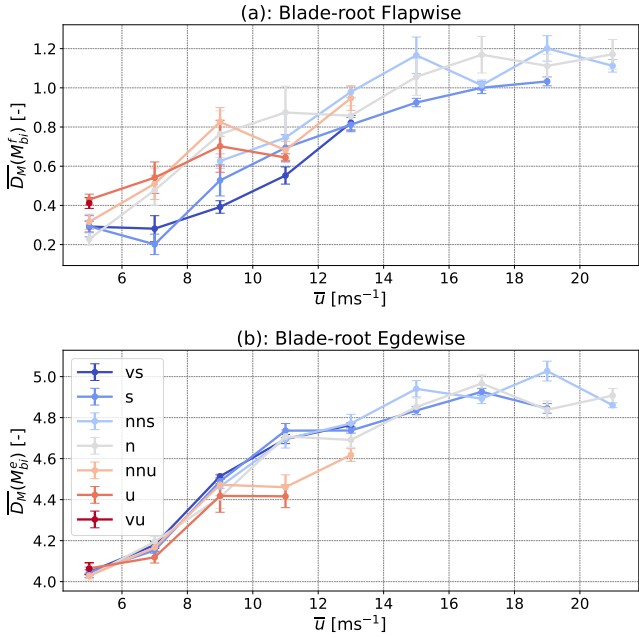

**Figure 7.** The mean damage equivalent moments at the root of blade 1 in the (a) flapwise and (b) edgewise directions as functions of wind speed and stability.

height. Stable conditions show converse behaviour such that the two drivers of fatigue loads tend to nullify each other in Fig. 7. Moreover, as pitch control is activated at wind speeds above rated, the blade pitches out of the mean wind. Hence the flapwise loads become less sensitive to turbulence and shear becomes relatively dominant. This causes stable conditions to be slightly

more detrimental at high wind speeds. In case of the edgewise loads on the blade, as they are mainly driven by gravity, no dependence on stability or turbulence is observed. Thus, we exclude them from the one-to-one comparisons presented in the next section. The side-side loads on the tower bottom are also excluded as they are not only dependent on the inflow but also on a range of different factors such as mass and pitch imbalances on the blades. It is difficult to replicate these in the simulation environment. Hence, attributing any disagreement between measured and simulated loads to differences in the inflow cannot be

achieved with a sufficient degree of confidence. On the other hand, the tower fore-aft and blade flapwise loads are significantly more sensitive to the inflow. Thus, our one-to-one comparison is limited to these two load channels.

While we present an analysis of load measurements, Sathe et al. (2013) showed the effect of atmospheric stability on fatigue loads in a simulation environment. However, our results appear to be remarkably similar which indicates a fundamental relationship between atmospheric stability and fatigue loads on a pitch controlled horizontal axis wind turbine.



## 3.2 One-to-one comparisons

The difference between damage equivalent moments in the simulations and measurements are quantified using the z-score, $z$, defined as follows:

$$z = \frac{\overline{A} - \overline{B}}{\sqrt{\sigma_{\overline{A}}^2 + \sigma_{\overline{B}}^2}}, \tag{16}$$

where,

$$\sigma_{\overline{A}} = \frac{\sigma_A}{N_A}. \tag{17}$$

$A$ and $B$ refer to the measured and simulated quantities respectively. $N_A$ is the number of 10 min samples whose values can be found in Table 3 while $N_B = 7$ which corresponds to the number of seeds used in each wind speed and stability bin. Note that when $z = 1$, $\overline{A}$ and $\overline{B}$ are $1\sigma$ apart, where:

$$\sigma = \sqrt{\sigma_{\overline{A}}^2 + \sigma_{\overline{B}}^2}. \tag{18}$$

Furthermore, a positive value indicates that fatigue loads are overestimated in the simulation while a negative value indicates an underestimation. The z-scores for $\overline{D_M}(M_{tb}^{fa})$ are presented in Table 4.

Figure 8 and Table 4 indicate that the tower-bottom damage equivalent moments are overestimated when inflow turbulence is modelled with the Mann model and the model parameters are derived by fitting the model to measurements of the auto-spectra. In most instances, the overestimation is fairly low and the loads differ by less than $3\sigma$. However, at certain wind speeds under near-neutral stable, stable, very stable and unstable conditions, the loads are overestimated by more than $5\sigma$. We found that in the bins with near-neutral, stable and very stable conditions, the vertical coherence in the $u$ component predicted by the model was higher than the measurements. Note that the coherence, $\gamma_{uu}^2$, is defined as:

$$\gamma_{uu}^2(f, \Delta_y, \Delta_z) = \frac{|\chi_{uu}(f, \Delta_y, \Delta_z)|^2}{S_u(f, 0, 0) S_u(f, \Delta_y, \Delta_z)}, \tag{19}$$

where, $\Delta_y$ and $\Delta_z$ are the lateral and vertical separations respectively and $\chi_{uu}$ is the cross spectrum between $u(t, 0, 0)$ and $u(t, \Delta_y, \Delta_z)$. Using data from the sonic anemometer at 155 m and the cup anemometer and wind vane at 45 m, we found measurements of $\gamma_{uu}^2(f, 0, 110)$ which were compared to the coherences from the turbulence model. In the frequencies between $10^{-3}$ Hz to $10^{-2}$ Hz, the model overestimated the coherence by up to 174%. We suspect that the lateral coherence is also overestimated in these cases. Nybø et al. (2021a) showed that coherence in the frequency range between $10^{-3}$ Hz to $10^{-2}$ Hz has a significant impact on the fatigue loads experienced by the towers of multi-megawatt turbines. Regarding the case with unstable conditions at 11 ms$^{-1}$, the measurements of the auto-spectra show that the variance in the $w$ component is higher than that in the $u$ and $v$ components at frequencies below $10^{-2}$ Hz. Such observations have been made before under strongly convective conditions (Syed, 2024) at an altitude of 200 m. However, the Mann model is based on the assumption that $\sigma_u > \sigma_v > \sigma_w$, which is true in case of shear-driven turbulence but invalid for some unstable conditions, especially for altitudes





**Figure 8.** The damage equivalent moment at the bottom of the tower in the fore-aft direction as measured on the prototype (in black) and simulated (in red) in the aeroelastic solver by assimilating the measurements of inflow via spectral fitting with the Mann turbulence model and a power law profile for the vertical variation in the wind speed.

above 100 m. Thus, fitting the model to the observations of the auto-spectra results in a relatively poor fit which we believe
295 causes the large mismatch between simulated and measured loads on the tower (and the blade). The one-to-one comparison of tower-top moments in the fore-aft direction were found to be identical to the tower-bottom.

The one-to-one comparison of the damage equivalent moments on the blade in the flapwise direction is shown in Fig. 9 while the z-scores are displayed in Table 5. In this case as well, the difference in loads is often less than $3\sigma$. At mean wind speeds of 9, 11 and 13 ms$^{-1}$, the damage equivalent moments under very stable and stable conditions are overestimated by
300 more than $11\sigma$. We find that the cause for this discrepancy is the method of assimilating the measured vertical shear into the simulation environment. The power law profile of Eq. (8) was fitted to measurements at and below hub-height and the shear





| $\overline{u}$-bin centre [ms$^{-1}$] | vs | s | nns | n | nnu | u | vu |
|---|---|---|---|---|---|---|---|
| 5 | 3.48 | -0.56 | - | 0.30 | 0.73 | 3.11 | 2.41 |
| 7 | 0.66 | 2.94 | - | 1.97 | -0.92 | 2.05 | - |
| 9 | 8.62 | 4.33 | -1.67 | -0.93 | -0.43 | -0.68 | - |
| 11 | 9.14 | -0.19 | -1.21 | 1.48 | 2.76 | 5.37 | - |
| 13 | -1.10 | 4.06 | 7.54 | 2.72 | 3.66 | - | - |
| 15 | - | 4.68 | 1.45 | 3.51 | - | - | - |
| 17 | - | 7.70 | 10.71 | 2.34 | - | - | - |
| 19 | - | 22.18 | 12.87 | 3.48 | - | - | - |
| 21 | - | - | 12.52 | 9.88 | - | - | - |

**Table 4.** The z-scores for the damage equivalent moments on the bottom of the tower in the fore-aft direction.

| $\overline{u}$-bin centre [ms$^{-1}$] | vs | s | nns | n | nnu | u | vu |
|---|---|---|---|---|---|---|---|
| 5 | 0.63 | -1.41 | - | 0.75 | -2.01 | 2.05 | 0.07 |
| 7 | 1.46 | 2.37 | - | 1.99 | -2.01 | 0.79 | - |
| 9 | 23.24 | 9.35 | -2.31 | -0.03 | 1.34 | 0.60 | - |
| 11 | 16.22 | 11.32 | 2.87 | 1.71 | 6.94 | 8.02 | - |
| 13 | 7.86 | 11.65 | 7.29 | 5.39 | 1.09 | - | - |
| 15 | - | 10.60 | 2.23 | 1.58 | - | - | - |
| 17 | - | 3.36 | 2.95 | -2.28 | - | - | - |
| 19 | - | -2.33 | -0.07 | -2.50 | - | - | - |
| 21 | - | - | -2.66 | -1.86 | - | - | - |

**Table 5.** The z-scores for the damage equivalent moments on the root of the blade in the flapwise direction.

coefficient was assumed to be independent of height. However, under stable conditions, the boundary layer height can be as low as 300 m. Thus, the vertical profile of wind speed between hub-height (163 m) and rotor-top (281 m) is not necessarily the same as between rotor-bottom (45 m) and hub height. By analysing the vertical wind profile measured from a nacelle lidar, we found that the power law overestimates the mean wind speeds above hub-height by 1 to 2 ms$^{-1}$. Consequently, the blade-root bending moment had a larger range in the simulations and the damage equivalent moments were overestimated.

## 4 Discussion and conclusions

This study aimed to evaluate the accuracy of load simulations in which inflow turbulence is prescribed using the Mann model (Mann, 1994), by comparing the resulting fatigue loads with measurements from a 15 MW wind turbine prototype. Such measurements are rarely available or published, making this analysis a valuable contribution. In addition, the study provides

**Blade-root Flapwise**

**(a): vs**

**(b): s**

**(c): nns**

**(d): n**

**(e): nnu**

**(f): u**

**Figure 9.** The damage equivalent moment at the root of the blade in the flapwise direction as measured on the prototype (in black) and simulated (in red) in the aeroelastic solver by assimilating the measurements of inflow via spectral fitting with the Mann turbulence model and a power law profile for the vertical variation in the wind speed.



insights into the influence of atmospheric stability on the fatigue damage accumulated by turbine towers and blades. The following conclusions can be drawn:

1. Analysis of the 1 Hz damage equivalent moment measured on the prototype demonstrated that atmospheric stability has a substantial effect on tower fatigue loads, while its impact on blade loads is less pronounced. These findings are consistent with Sathe et al. (2013), who additionally showed that the IEC recommendation can lead to estimations of lifetime fatigue damage that are up to 96% higher than those obtained when accounting for the joint distribution of mean wind speed and stability at a given site. Our results therefore reinforce the case for including atmospheric stability in fatigue load assessments as a means of reducing over-design and associated costs. The alternative to the IEC approach is a site-specific fatigue load estimation, but current implementations typically assume neutral conditions to be dominant. While this assumption holds for many onshore sites, it may not be valid offshore or in coastal environments. Insufficient consideration of atmospheric stability at such sites could lead to underestimation of lifetime fatigue damage on wind turbine towers.

2. A one-to-one comparison of measured and simulated fatigue loads, where the inflow was assimilated by fitting turbulence spectra to the Mann model, showed good overall agreement. Unlike most previous studies, our simulations used the same controller as the prototype, ensuring a more realistic comparison. The damage equivalent moments from simulations were generally higher than measurements, but the margin of overestimation was small enough to be acceptable for engineering purposes. Improved inflow assimilation, for example through turbulence boxes constrained by nacelle lidar measurements, could further reduce these discrepancies. Alternatively, these differences could stem from uncertainties in the aerodynamic or structural properties of the simulated turbine, which are outside the scope of this study. Overall, we conclude that the Mann turbulence model is suitable for aeroelastic simulations of multi-megawatt wind turbines. Known limitations of the model, such as its assumption of Gaussian small-scale statistics and homogeneous turbulence, do not appear to significantly affect fatigue load predictions for tower and blade loads of a stand-alone turbine. Although the model was originally formulated for surface-layer, shear-driven turbulence, it successfully captures the key turbulence characteristics (length scale, turbulence intensity, and coherence) most relevant for turbine loads. However, it is uncertain whether these results can be extrapolated to yaw and tilt moments on the main shaft.

3. In some one-to-one comparisons, significant overestimations of tower and blade fatigue loads were observed under certain stable conditions. For the tower, this was attributed to the Mann model overestimating vertical coherence in the along-wind component, while blade loads were overestimated due to an inaccurate shear profile. The physical mechanisms leading to the low measured coherence remain unclear, but future work could investigate whether the model of Chougule et al. (2018) is more accurate under some stable conditions. Furthermore, better representation of the shear profile above hub height in the simulations could be achieved through nacelle- or ground-based lidars capable of measuring up to the rotor top, or by adopting an extended boundary layer wind profile (Gryning et al., 2007). Accurate assimilation of the shear profile can also be important for power curve verification (Wharton and Lundquist, 2012).



Subsequent research can be directed towards conducting one-to-one comparisons for wind turbines operating within wind
farms. As the Mann model is assumed to describe atmospheric turbulence in time-domain simulations of wake meandering
(Larsen et al., 2008), such studies would help determine whether the model remains valid in this context or extended for-
mulations (Syed and Mann, 2024) are preferable. In addition, one-to-one comparisons could be performed for floating wind
turbines, with inflow characterized using the Mann model, the Kaimal model (Kaimal et al., 1972) with exponential coherence
(Davenport, 1961), or Large Eddy Simulations (Doubrawa et al., 2024).

*Data availability.*  For reasons of confidentiality, the data analysed in this study is not publicly available.

*Author contributions.*  AP and JM conceptualized the research question. AP performed the analysis with assistance from JM, MS, KZ and
KR. AP also wrote the first version of the manuscript which was subsequently reviewed and edited by all co-authors.

*Competing interests.*  JM is a chief editor of the *Wind Energy Science* journal. KZ and KR are employed by Vestas Wind System A/S. The
authors have no other competing interests to declare.

*Acknowledgements.*  This project has received funding from the European Union's Horizon Europe research and innovation program under
the Marie Skłodowska-Curie grant agreement No 101119550 (AptWind). The authors would like to thank Stefan Wolff from Vestas for
assistance in running and debugging the aeroelastic simulations. We are also grateful to Ahmad Khamas from Vestas for helping us navigate
and use the measurement data from the wind turbine prototype.



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
