# Peer review of "Comparison of measured and simulated fatigue loads on a multi-megawatt wind turbine"

_Wind Energy Science, 2025_

## Referee Comment (RC2)

**Comparison of measured and simulated fatigue loads on a multi-megawatt wind turbine**

January 26, 2026

**General comments:**

The manuscript entitled "**Comparison of measured and simulated fatigue loads on a multi-megawatt wind turbine**" by Patel et al. presents a valuable validation and measurement case to address the effect of atmospheric stability in turbulence models on fatigue loads on large, possibly one of the newest, offshore wind turbine. The paper presents full-scale measurement data which is used to derived empirical variation of parameters for several cases of non-neutral atmospheric stability. Considering the realistic validation case and the valuable comparison directly using full-scale measurement data to the in-house simulation results, the paper is suitable for the journal.

However, several points are still need to be addressed in the paper. I recommend major revision to address the points detailed in "**specific comments**" and "**technical corrections**" below.

**Specific comments:**

1. Line 24 - Could you point out which section in Veers et al. (2023) that support the statement? Do you mean the second last paragraph before Section 3.3 in Veers et al. (2023) [1]? If yes, please reference more accurately.

2. Line 40 - I assume "which method" here refers to Mann, Kaimal models and wind fields from LES. It would be easier to mention them as "which inflow assumptions" or "which inflow models".

3. Line 57 - It is unclear which "two methods" meant here. Does this refer to the method discussed in Rinker (2022) [2]? Could you elaborate/revise for accuracy?

4. Line 58 - I couldn't find the information that is related to the statement "...*found that the uncertainties in the load estimates from the two methods were similar and occasionally lower when using the lidar*" in Dimitrov et al. (2024) [3]. Perhaps, do you mean the other paper by the same author, Dimitrov and Natarajan (2017) [4]? If yes, please correct the reference.

5. The paper's contribution on using real-scale/full-scale measurements should be highlighted. It would be helpful to add one or two sentence in the introduction how important the validation using full-scale measurement data is and the importance of full-scale measurement (in real scale and natural atmosphere). It can be placed before the organization of the paper paragraph.

6. Lines 65 and 66 - "..., *the relationship between fatigue loads on a wind turbine operating in the field and the atmospheric stability has previously not been documented in the literature.*". Do you mean specifically for the large, newest offshore wind turbine (15MW)? The work of Holtslag (2016) [5] provides the analysis of atmospherical stability on fatigue calculation, though on 5 MW.

7. Line 80 - "data signals" would perhaps be clearer instead of just "signals"

8. Lines 85 to 88: Please consider making this paragraph more concise. Redundancy has been detected, for example in the explanation of 'filtered dataset', when the authors may have

meant 'curated'. The paragraph may need to be reorganised, as the next paragraph (Line 89) explains the first steps of data processing, which would be more appropriate to be placed at the beginning.

9. Line 89 - is 5 Hz enough? Could you provide explanation the effect of this chosen sampling frequency?

10. Line 89 - "*Only those periods were selected for further analysis, ...*". The author focuses on the evaluation on a specific wind direction of 225° - 315°, which is one of the "curation" steps that they performed to build the data sets. But I suggest to previously give more information on the data availability. For example, the measurement campaign that is used lasts between 2023 to 2025, then later it is stated that in the summer 2024 there are instruments error, so there are no measured data. Such information are currently scattered in the paper. A short paragraph or small diagram/table to explain data availability might help the reader from which period of months and year the chosen data are.

11. Line 91 - Is the averaging of 10-minute? Clearly stating the "10-minute" might help to remind the reader.

12. Line 104 - Please detail which anemometer and which height is used for the mean wind speed, also directly in the sentence.

13. Line 107 to 109 - Which height do you compare to of the works Peña (2019) [6]? I assume based on the statement in these lines, it is compared to Figure 11(b) of Peña (2019)? Then, could you explain how is the effect of the comparison to different height, since the measured data in this paper is from sonic anemometer 155 m, and in Peña (2019), it is from 241 m.

14. Line 168 to 169 - A power-law fitting is used to characterize the mean wind speed profile (wind profile) along the height, and obtain exponent parameter $\alpha_s$. However, the power law is actually appropriate for neutral atmospheric condition. I expect fitting a power-law profile to only three wind speed measurement point (45 m, 105 m and 163 m) might be a simplification. I suggest to discuss the quality of fitting for the obtained $\alpha_s$, such as what is the $R^2$ range for the fitting used here to obtained the $\alpha_s$ resulting the Figure 5(a)?

15. Line 213 - Since the authors decided to anonymize the values, please explain more the impact to be expected from this anonymization.

16. Line 235 - It is stated that "*As seen in Fig. 3 and Fig. 5, unstable conditions are generally associated with higher turbulence intensities and larger length scales ...*". Which figure or data exactly that shows the effect of turbulence intensity? Do you mean indicated by the variance of spectra in Fig. 3? I expect, especially in lower wind speed, variation of turbulence intensities are present, especially along the height. I suggest to provide information from measurement on how is the turbulence intensities (in terms of ratio of velocity fluctuation to mean wind velocity, e.g. $I_u = u'/\bar{u}$) along the height related to each of the stability classes or classified by the wind speed bins. Or provide a figure that shows turbulence intensity as its own value based on stability classes. Especially that later, the authors discussed a lot of "turbulence intensity" in paragraph of Line 240.

17. Figure 7 - which wind speed height are used for the x-axis? 155 m?

18. Line 248 - Please add the value of rated wind speed (even if it's an estimation only), such as "At wind speeds below rated **(... m/s)**, "

19. Line 252 - Could you shortly mention the two drivers of fatigue loads in this line? Maybe shortly, inside a bracket (...)

20. Line 255 to 256 - It is stated that "*in case of the edgewise loads on the blade, as they are mainly driven by gravity, no dependence on stability or turbulence is observed.*" But in Figure 7(b) before rated wind speed (approximately), the unstable and very unstable show lower damage. How does this sentence then stay true?

21. Figure 9(d) - In neutral condition, the measured damage values on blade root (flapwise) are higher than simulated for larger wind speed (larger than 16 m/s). Could you address this in the paper?

22. Line 300 to 305 - Interesting!

**Technical corrections:**

1. Equation (2), Line 103 - I assume, the equation of friction velocity is 1-D simplifed, in the along-wind ($u$) wind direction. There should be however apostrophe (') for each velocity component $u$ and $w$, as it works with velocity fluctuations.

2. Table 2, column 2 - Please add the unit of Monin-Obukhov length, such as "(m)" for easier reading

3. Line 154 - Do you mean "...with the fit derived from Eq (**7**) ..."?

4. Figure 3 - Please consider to put the legend on Figure 3(a) instead of on 3(f)

5. Line 227 - Regarding the word "straddles", maybe use simpler word more appropriate to the context, such as "which **is around** the turbine rated wind speed"

**References**

[1] P. Veers et al. Grand challenges in the design, manufacture, and operation of future wind turbine systems *Wind Energy Science*, 8, 1071–1131, 2023.

[2] J. Rinker. Impact of rotor size on aeroelastic uncertainty with lidar-constrained turbulence, *Journal of Physics: Conference Series*, 2265, 032011, https://doi.org/10.1088/1742-6596/2265/3/032011, 2022.

[3] N. Dimitrov, M. Pedersen, and Á. Hannesdóttir. An open-source Python-based tool for Mann turbulence generation with constraints and non-Gaussian capabilities. *Journal of Physics: Conference Series*, 2767, 052058, 2024.

[4] N. K. Dimitrov and A. Natarajan. Application of simulated lidar scanning patterns to constrained Gaussian turbulence fields for load validation. *Wind Energy*, 20(1):79–95, https://doi.org/10.1088/1742-6596/2767/5/052058, 2017.

[5] Holtslag, M. C., Bierbooms, W. A. A. M., and van Bussel, G. J. W. Wind turbine fatigue loads as a function of atmospheric conditions offshore. *Wind Energy*, 19: 1917–1932. doi: 10.1002/we.1959, 2016.

[6] Peña, A. Østerild: A natural laboratory for atmospheric turbulence. *J. Renewable Sustainable Energy*, 11, 063302. https://doi.org/10.1063/1.5121486, 2019.